# Optimal percolation on multiplex networks

Saeed Osat[1,3], Ali Faqeeh[2] & Filippo Radicchi[2]

Optimal percolation is the problem of finding the minimal set of nodes whose removal from a network fragments the system into non-extensive disconnected clusters. The solution to this problem is important for strategies of immunization in disease spreading, and influence maximization in opinion dynamics. Optimal percolation has received considerable attention in the context of isolated networks. However, its generalization to multiplex networks has not yet been considered. Here we show that approximating the solution of the optimal percolation problem on a multiplex network with solutions valid for single-layer networks extracted from the multiplex may have serious consequences in the characterization of the true robustness of the system. We reach this conclusion by extending many of the methods for finding approximate solutions of the optimal percolation problem from single-layer to multiplex networks, and performing a systematic analysis on synthetic and real-world multiplex networks.

---

[1] Molecular Simulation Laboratory, Department of Physics, Faculty of Basic Sciences, Azarbaijan Shahid Madani University, Tabriz 53714-161, Iran. [2] Center for Complex Networks and Systems Research, School of Informatics and Computing, Indiana University, Bloomington, IN 47408, USA. [3] Quantum Complexity Science Initiative, Skolkovo Institute of Science and Technology, Skoltech Building 3, Moscow 143026, Russia. Correspondence and requests for materials should be addressed to F.R. (email: filiradi@indiana.edu)

A multiplex is a network in which nodes are connected through different types or flavors of pairwise edges[1–3]. A convenient way to think of a multiplex is as a collection of network layers, each representing a specific type of edges. Multiplex networks are genuine representations for several real-world systems, including social[4,5], and technological systems[6,7]. From a theoretical point of view, a common strategy to understand the role played by the co-existence of multiple network layers is based on a rather simple approach. Given a process and a multiplex network, one studies the process on the multiplex and on the single-layer projections of the multiplex (e.g., each of the individual layers, or the network obtained from aggregation of the layers). Recent research has demonstrated that ignoring the effective co-existence of different types of interactions in the study of a multiplex network may have dramatic consequences in the ability to model and predict properties of the system. Examples include dynamical processes, such as diffusion[8,9], epidemic spreading[10–13], synchronization[14], and controllability[15], as well as structural processes such as those typically framed in terms of percolation models[16–29].

The vast majority of the work on structural processes on multiplex networks have focused on ordinary percolation models where nodes (or edges) are considered either in a functional or in a non-functional state with homogenous probability[30]. In this paper, we shift the focus on the optimal version of the percolation process: we study the problem of identifying the smallest set of nodes in a multiplex network such that, if these nodes are removed, the network is fragmented into many disconnected clusters with non-extensive sizes. We refer to the nodes belonging to this minimal set as structural nodes (SNs) of the multiplex network. The solution of the optimal percolation problem has

direct applicability in the context of robustness, representing the cheapest way to dismantle a network[31–33]. The solution of the problem of optimal percolation is, however, important in other contexts, being equivalent to the best strategy of immunization to a spreading process, and also to the best strategy of seeding a network for some class of opinion dynamical models[34–37]. Despite its importance, optimal percolation has been introduced and considered in the framework of single-layer networks only recently[35,36]. Optimal percolation is a NP-complete problem[32]. Hence, on large networks, we can only use heuristic methods to find approximate solutions. Most of the research activity on this topic has indeed focused on the development of greedy algorithms[31–33,35].

Here we consider the generalization of optimal percolation to multiplex networks. Our generalization consists in the redefinition of the problem in terms of mutual connectedness[16]. To this end, we reframe several algorithms for optimal percolation in single-layer networks to obtain methods that consider the multiplex structure of networks as well. Basically all the algorithms we use provide coherent solutions to the problem, finding sets of SNs that are almost identical. Our main focus, however, is not on the development of new algorithms, but on understanding the consequences that arise from neglecting the multiplex nature of a network under an optimal percolation process. We compare the actual solution of the optimal percolation problem in a multiplex network with the solutions to the same problem for single-layer networks extracted from the multiplex system. We show that "forgetting" about the presence of multiple layers can be potentially dangerous, leading to the overestimation of the true robustness of the system mostly due to the identification of a very high number of false SNs. We reach this conclusion with a systematic analysis of both synthetic and real-world multiplex networks.

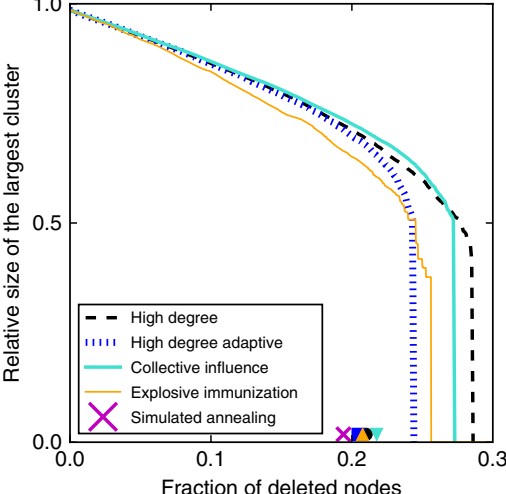

**Fig. 1** Performance of different algorithms aimed at solving the optimal percolation problem. We consider a multiplex network with $N = 10,000$ nodes. The multiplex is composed of two network layers generated independently according to the Erdös–Rényi model with average degree $\langle k \rangle = 5$. Each curve represents the relative size of the GMCC as a function of the relative number of nodes inserted in the set of SNs, thus removed from the multiplex. Colored markers indicate the effective fraction of nodes left in the set of SNs after a greedy post-processing technique is applied to the set found by the corresponding algorithm. The purple cross identifies instead the size of the set of SNs found through Simulated Annealing optimization. Please note that the ordinate value of the markers has no meaning; in all cases, the relative size of the largest cluster is smaller than $N^{1/2}$. Details on the implementation of the various algorithms are provided in Supplementary Notes 1, 2

## Results

**Identifying structural nodes in multiplex networks**. We consider a multiplex network composed of $N$ nodes arranged in two layers. Each layer is an undirected and unweighted network. Connections of the two layers are encoded in the adjacency matrices. **A** and **B**. The generic element $A_{ij} = A_{ji} = 1$ if nodes $i$ and $j$ are connected in the first layer, whereas $A_{ij} = A_{ji} = 0$, otherwise. The same definition applies to the second layer, and thus to the matrix **B**. The aggregated network obtained from the superposition of the two layers is characterized by the adjacency matrix **C**, with generic elements $C_{ij} = A_{ij} + B_{ij} - A_{ij}B_{ij}$. We focus our attention on clusters of mutually connected nodes[16]: two nodes in a multiplex network are mutually connected, and thus part of the same cluster of mutually connected nodes, only if they are connected by at least a path, composed of nodes within the same cluster, in every layer of the system. In particular, we focus our attention on the largest among these cluster, usually referred to as the giant mutually connected cluster (GMCC). Our goal is to find the minimal set of nodes such that, if removed from the multiplex, no mutual cluster with a size greater than $N^{1/2}$ is found in the network. This is a common prescription, yet not the only one possible, to ensure that all clusters have non-extensive sizes in systems with a finite number of elements[35]. Whenever we consider single-layer networks, the above prescription applies to the single-layer clusters in the same exact way.

We generalize most of the algorithms devised to find approximate solutions to the optimal percolation problem in single-layer networks to multiplex networks[31–33,35,36]. Details on the implementation of the various methods are provided in the Supplementary Note 1. We stress that the generalization of these methods is not trivial at all. For instance, most of the greedy

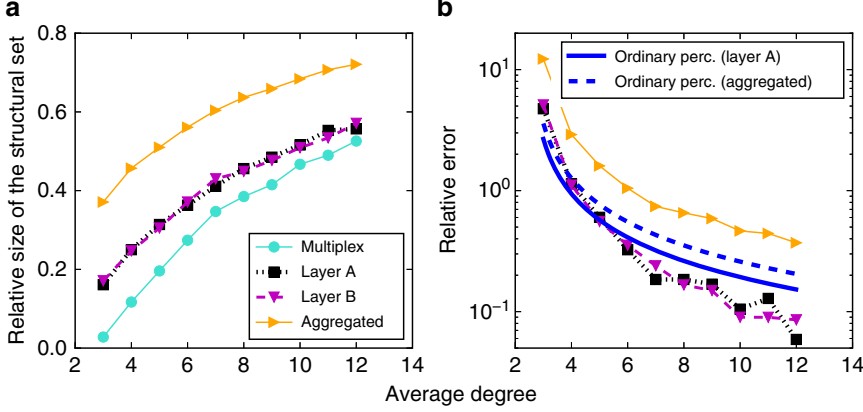

**Fig. 2** Optimal percolation problem on synthetic multiplex networks. **a** We consider multiplex networks with $N = 1,000$ and layers generated independently according to the Erdös−Rényi model with average degree $\langle k \rangle$. We estimate the relative size of the set of SNs on the multiplex as a function of $\langle k \rangle$ (turquoise circles), and compare it with the same quantity but estimated on the individual layers (black squares and purple triangles) or the aggregated (orange triangles). **b** The relative errors of single-layer estimates of the size of the structural set with respect to the ground-truth value provided by the multiplex estimate. Colors and symbols are the same as those used in **a**. The blue curves with no markers represent instead the theoretically expected behavior for an ordinary percolation process[16]

methods use node degrees as crucial ingredients to calculate and assign scores to each of the nodes, and then remove nodes with respect to their scores. In a multiplex network, however, a node has multiple degree values, one for every layer. In this respect, it is not clear what is the most effective way of combining these numbers to assign a single score to a node: they may be summed, thus obtaining a number approximately equal to the degree of the node in the aggregated network derived from the multiplex, but also multiplied, or combined in more complicated ways. We find that the results of the various algorithms are not particularly sensitive to this choice, provided that the simple but effective post-processing technique considered in refs. [31–33] is applied to the set of SNs found by a given method. In Fig. 1, for example, we show the performance of several greedy algorithms when applied to a multiplex network composed of two layers generated independently according to the Erdős−Rényi (ER) model. Although the mere application of an algorithm may lead to different estimates of the size of the set of SNs, if we greedily remove from these sets the nodes that do not increase the size of the GMCC to the predefined sub-linear threshold ($N^{1/2}$)[31–33] (Supplementary Note 2), the sets obtained after this post-processing technique have almost identical sizes (Supplementary Figs. 1–4).

As Fig. 1 clearly shows, the best results, in the sense that the size of the set of SNs is minimal, is found with a simulated annealing (SA) optimization strategy[32] (see details in the Supplementary Note 1). The fact that the SA method is outperforming score-based algorithms is not surprising. SA actually represents one of the best strategies that one can apply in hard-optimization tasks. In our case, it provides us with a reasonable upper bound on the size of the set of SNs that can be identified in a multiplex network. The second advantage of SA in our context is that it does not rely on ambiguous definitions of ingredients (e.g., node degree). Despite its better performance, SA has a serious drawback in terms of computational speed. As a matter of fact, the algorithm can be applied only to multiplex networks with moderate sizes. As here we are interested in understanding properties of the optimal percolation problem in multiplex networks, the analysis presented in the main text of the paper is entirely based on results obtained through SA optimization. This provides us with a solid ground to support our statements. Extending the analysis of score-based algorithms to larger multiplex networks leads to qualitatively similar results (Supplementary Note 3, Supplementary Figs. 5–8).

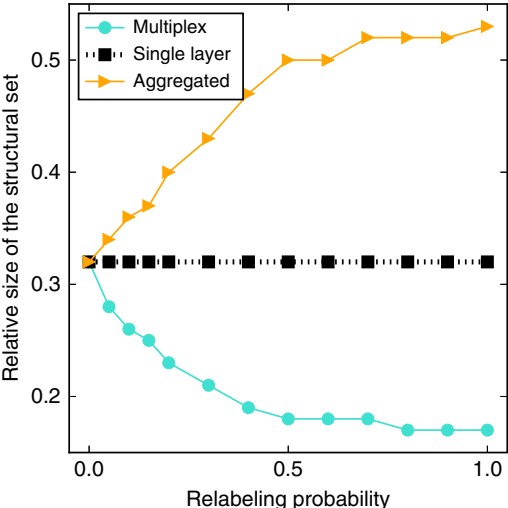

**Fig. 3** Dependence of the size of the structural set on edge overlap and interlayer degree–degree correlation. We consider multiplex networks, in which, initially, both layers are a copy of a random network generated according to the Erdős−Rényi model with $N = 1,000$ nodes and average degree $\langle k \rangle = 5$. Then, in one of the layers, each node is selected to switch its label with another randomly chosen node with a certain probability $\alpha$. We determine, as a function of $\alpha$, the mean value of the relative size of the set of SNs over 100 realizations of the SA algorithm on the multiplex network

**The size of the set of structural nodes**. We consider the relative size of the set of SNs, denoted by $q$, for a multiplex composed of two independently fabricated ER network layers as a function of their average degree $\langle k \rangle$. We compare the results obtained applying the SA algorithm to the multiplex, namely $q_M$, with those obtained using SA on the individual layers, i.e., $q_A$ and $q_B$, or the aggregated network generated from the superposition of the two layers, i.e., $q_S$. By definition, we expect that $q_M \leq q_A \simeq q_B \leq q_S$. What we do not know, however, is how bad/good are the measures $q_A$, $q_B$ and $q_S$ in the prediction of the effective robustness of the multiplex $q_M$. For ordinary random percolation on ER multiplex networks with negligible overlap, we know that $q_M \simeq 1 - 2.4554/\langle k \rangle$[16],

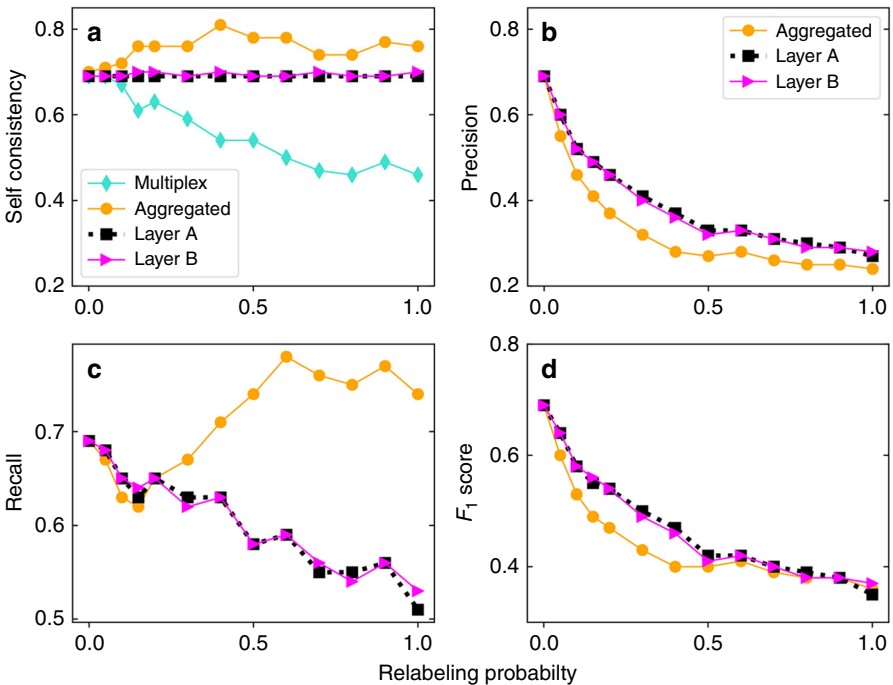

**Fig. 4** Comparison between structural sets obtained for different network representations. We consider the multiplex networks described in Fig. 3 and the sets of SNs found for the multiplex and single-layer based representations of these networks. **a** As the set of SNs found in different instances of the optimization algorithm are different from each other, we first quantify the self-consistency of those solutions across 100 independent runs of the SA algorithm. We then assume that the multiplex representation provides the ground-truth classification of the nodes. We compare the results of the other representations with the ground truth by measuring their precision **b**, their sensitivity or recall **c**, and their $F_1$ score **d**

$q_A \simeq q_B \simeq 1 - 1/\langle k \rangle$, and $q_S \simeq 1 - 1/(2\langle k \rangle)$[38]. Relative errors are therefore $\varepsilon_A \simeq \varepsilon_B \simeq (2.4554 - 1)/(\langle k \rangle - 2.4554)$, and $\varepsilon_S \simeq (2.4554 - 1/2)/(\langle k \rangle - 2.4554)$. We find that the relative error for optimal percolation behaves more or less in the same way as that of ordinary percolation (Fig. 2b), noting that, as $\langle k \rangle$ is increased, the decrease in the relative error associated with the individual layers is slightly faster than what expected for ordinary percolation. The relative error associated with the aggregated network is larger than the one expected from the theory of ordinary percolation. As shown in Fig. 2a, for sufficiently large $\langle k \rangle$, dismantling an ER multiplex network is almost as hard as dismantling any of its constituent layers.

**Edge overlap and degree correlations**. Next, we test the role played by edge overlap and layer-to-layer degree correlation in the optimal percolation problem. These are the ingredients that dramatically change the nature of the ordinary percolation transition in multiplex networks[26,39–43]. In Fig. 3, we report the results of a simple analysis. We take advantage of the model introduced in ref. [44]. This is one of the simplest models able to tune a system from a multiplex to a simplex topology. The system is composed of two identical network layers. Nodes in one of the two layers are relabeled with a certain probability $\alpha$. For $\alpha = 0$, multiplex, aggregated network and single-layer graphs are all identical. For $\alpha = 1$, the networks are analogous to those considered in the previous section. We note that this model does not allow to disentangle the role played by edge overlap among layers and the one played by the correlation of node degrees. For $\alpha = 0$, edge overlap amounts to 100%, and there is a one-to-one match between the degree of a node in one layer and its degree in the other layer. As $\alpha$ increases, both edge overlap and degree correlation decrease simultaneously. As it is apparent from the results of Fig. 3, the system reaches the multiplex regime for very small values of $\alpha$, in the sense that the relative size of the set of SNs

deviates instantly from its value for $\alpha = 0$. This is in line with what already found in the context of ordinary percolation processes in multiplex networks: as soon as there is a finite fraction of edges that are not shared by the two layers, the system behaves exactly as a multiplex[26,39–43].

**Accuracy and sensitivity**. So far, we focused our attention only on the size of the set of SNs. We neglected, however, any analysis regarding the identity of the nodes that actually compose this set. To proceed with such an analysis, we note that different runs of the SA algorithm (or any algorithm with stochastic features) generally produce slightly different sets of SNs, even if they all have almost identical sizes. The issue is not related to the optimization technique, rather to the existence of degenerate solutions to the problem. In this respect, we work with the quantities $p_i$, each of which describes the probability that a node $i$ appears in the set of SNs in a realization of the detection method (here, the SA algorithm). This treatment takes into account the fact that a node may belong to the set of SNs in a number of realizations of the detection method and may be absent from this set in some other realizations.

We define self-consistency of a SN-detection method as $S = \left[\sum_i p_i^2\right] / \left[\sum_i p_i\right]$, which describes the ratio of the expected overlap between two SNs obtained from two independent realizations of the detection method to the expected size of the SN. If the set of SNs is identical across different runs, then $S = 1$. The minimal value we can observe is $S = Q/N$, assuming that the size of the structural set is equal to $Q$ in all runs, but nodes belonging to this set are changing all the times, so that for every node $i$ we have $p_i = Q/N$. As reported in Fig. 4a, self-consistency $S$ assumes high values for single-layer representations of the network, even for synthetic multiplex networks. On the other hand, $S$ decreases significantly as the overlap and interlayer degree correlations decrease (Fig. 4a). Low $S$ values for

**Table 1 Optimal percolation on real multiplex networks**

| Network | Layers | N | Multiplex | | Single layers | | | | | | | | Aggregate | | | |
|---|---|---|---|---|---|---|---|---|---|---|---|---|---|---|---|---|
| | | | $q_M$ | $S$ | $q_A$ | $P_A$ | $R_A$ | $F_1^{(A)}$ | $q_B$ | $P_B$ | $R_B$ | $F_1^{(B)}$ | $q_S$ | $P_S$ | $R_S$ | $F_1^{(S)}$ |
| Air transportation[26] | American Air.—Delta | 84 | 0.12 | 0.85 | 0.14 | 0.58 | 0.70 | 0.63 | 0.32 | 0.29 | 0.79 | 0.42 | 0.35 | 0.32 | 0.92 | 0.47 |
| | American Air.—United | 73 | 0.10 | 0.99 | 0.16 | 0.32 | 0.52 | 0.40 | 0.14 | 0.68 | 1.00 | 0.81 | 0.25 | 0.39 | 1.00 | 0.56 |
| | United—Delta | 82 | 0.10 | 1.00 | 0.27 | 0.23 | 0.62 | 0.34 | 0.12 | 0.80 | 1.00 | 0.89 | 0.33 | 0.30 | 1.00 | 0.46 |
| C. Elegance[47,48] | Electric—Chem. Mon. | 238 | 0.09 | 0.69 | 0.16 | 0.41 | 0.71 | 0.52 | 0.26 | 0.22 | 0.60 | 0.32 | 0.35 | 0.21 | 0.79 | 0.33 |
| | Electric—Chem. Pol. | 252 | 0.12 | 0.79 | 0.15 | 0.50 | 0.63 | 0.56 | 0.39 | 0.24 | 0.78 | 0.37 | 0.45 | 0.22 | 0.82 | 0.35 |
| | Chem. Mon—Chem. Pol. | 259 | 0.25 | 0.82 | 0.28 | 0.69 | 0.77 | 0.73 | 0.39 | 0.51 | 0.79 | 0.62 | 0.42 | 0.48 | 0.80 | 0.60 |
| Arxiv[49] | physics.data-an—cond-mat.dis-nn | 1400 | 0.05 | 0.78 | 0.10 | 0.38 | 0.77 | 0.51 | 0.07 | 0.55 | 0.75 | 0.63 | 0.13 | 0.31 | 0.81 | 0.45 |
| | physics.data-an—cond-mat.stat-mech | 709 | 0.03 | 0.73 | 0.08 | 0.23 | 0.67 | 0.34 | 0.03 | 0.64 | 0.72 | 0.68 | 0.09 | 0.22 | 0.74 | 0.34 |
| | cond-mat.dis-nn—cond-mat.stat-mech | 499 | 0.02 | 0.50 | 0.06 | 0.13 | 0.46 | 0.20 | 0.04 | 0.23 | 0.51 | 0.32 | 0.09 | 0.13 | 0.65 | 0.22 |
| Drosophila M.[50,51] | Direct—Supp. Gen. | 676 | 0.01 | 0.62 | 0.07 | 0.12 | 0.60 | 0.20 | 0.11 | 0.09 | 0.64 | 0.16 | 0.19 | 0.07 | 0.87 | 0.13 |
| | Direct—Add. Gen. | 626 | 0.01 | 0.81 | 0.07 | 0.06 | 0.64 | 0.11 | 0.09 | 0.05 | 0.59 | 0.09 | 0.16 | 0.04 | 0.87 | 0.08 |
| | Supp. Gen.—Add. Gen. | 557 | 0.09 | 0.82 | 0.14 | 0.44 | 0.74 | 0.55 | 0.12 | 0.50 | 0.70 | 0.58 | 0.20 | 0.35 | 0.80 | 0.49 |
| Homo S.[48,50] | Direct—Supp. Gen. | 4465 | 0.05 | 0.72 | 0.16 | 0.20 | 0.73 | 0.31 | 0.13 | 0.23 | 0.64 | 0.34 | 0.27 | 0.15 | 0.89 | 0.26 |
| | Physical—Supp. Gen. | 5202 | 0.05 | 0.75 | 0.15 | 0.23 | 0.77 | 0.35 | 0.13 | 0.22 | 0.63 | 0.33 | 0.26 | 0.16 | 0.90 | 0.27 |

From left to right we report the following information. The first three columns contain the name of the system, the identity of the layers, and the number of nodes of the network. The fourth and fifth columns are results obtained from the optimal percolation problem studied on the multiplex network, and contain information about the relative size $q_M$, and self-consistency metric $S$ of the set of SNs. Then, we report results obtained for the first single-layer network of the multiplex, namely the fraction $q_A$ of nodes in the structural set, the precision $P_A$, the recall $R_A$, and the $F_1$ score of the set of SNs of the first layer. The next four columns are identical to those, but refer to the second layer. Finally, the four rightmost columns contain information about the fraction $q_S$ of nodes in the structural set, $P_S$ precision, $R_S$ recall, and the $F_1$ score of the set of SNs for the aggregated network obtained from the superposition of the two layers. All results have been obtained with 100 independent instances of the SA optimization algorithm

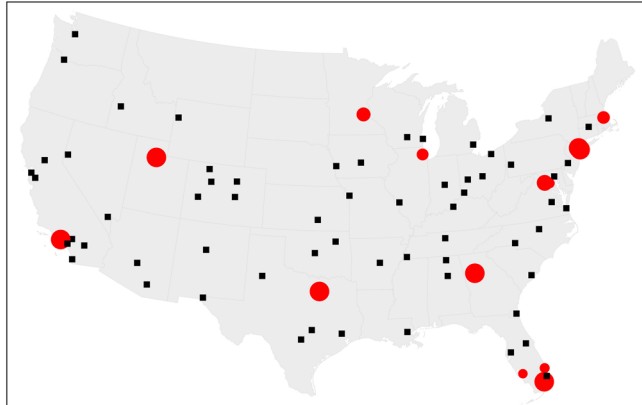

**Fig. 5** Optimal percolation on multiplex transportation networks. We consider the multiplex network of US domestic flights operated in January 2014 by American Airlines and Delta. Red circles represent nodes that were identified as members of the set of structural nodes in different realizations of the optimal percolation on the multiplex representation of the network. The size of each circle is proportional to the probability of finding that node in the set of SNs. All other airports in the multiplex are represented as black squares. Interestingly, not all the 14 structural nodes match the top 14 busiest "hubs" (https://en.wikipedia.org/wiki/List_of_the_busiest_airports_in_the_United_States), nor the probabilities follow the same order as the flight traffic of these airports. The results have been obtained with 100 independent instances of the SA optimization algorithm

multiplexes with small overlap and correlation together with the small sizes of their set of SNs (Fig. 2) suggest that in such networks many slightly different SN sets may exist.

Next, we turn our attention on quantifying how the sets of SNs identified in single-layer or aggregated networks are representative of the ground-truth sets found on multiplex networks. We denote by $p_i$ and $w_i$ the probability that node $i$ is found within the set of SNs of, respectively, a multiplex network (ground truth) and a specific single-layer representation of that multiplex. To compare the sets represented by $w_i$ with the ground-truth sets, we adopt three standard metrics in information retrieval[45,46], namely precision, recall and the Van Rijsbergen's $F_1$ score. Precision is defined as $P = [\sum_i p_i w_i]/[\sum_i w_i]$, i.e., the ratio of the expected number of correctly detected SNs to the expected total number of detected SNs. Recall is defined as $R = [\sum_i p_i w_i]/[\sum_i p_i]$, i.e., the ratio of the expected number of correctly detected SNs to the expected number of actual SNs of the multiplex. We note that the self-consistency we previously defined corresponds to precision and recall of the ground-truth set with respect to itself, thus providing a base line for the interpretation of the results. The $F_1$ score defined as $F_1 = (2)/(1/P + 1/R)$ provides a balanced measure in terms of $P$ and $R$. As Fig. 4b shows, $P$ deteriorates as the edge overlap and interlayer degree correlation decrease. In particular, when overlap and correlation between the layers of the multiplex network are not large, precision values for the sets of SNs identified in single layers or in the superposition of the layers are quite small ($P \simeq 0.3$), even smaller than the ratio of the $q_M$ of the multiplex to the $q$ of any of these sets (Fig. 3). This means that, when the multiplex nature of the system is neglected, two systematic errors are committed. First, the number of SNs is greatly overestimated; second, a significant number of the true SNs of the multiplex are not identified. The quantity $R$, on the other hand, behaves differently for single-layer and aggregated networks (Fig. 4c). In single layers, we see that $R$ systematically decreases as the relabeling probability increases. The structural set of nodes obtained on the superposition of the layers instead provides large values of $R$. This is not due to a good performance rather to the fact that the set of SNs identified on the aggregated network is

very large (Fig. 3), and it is further supported by the results of Fig. 4c, d, where large $R$ values do not correspond to high $F_1$ scores.

**Real-world multiplex networks**. In Table 1, we present summary statistics of the solution of the optimal percolation problem studied on several real-world multiplex networks generated from empirical data. For most of these networks, we find high values of self-consistency among solutions. This implies that there is a certain small group of nodes that have a major importance in the robustness of such real-world networks to the optimal percolation process. For most of the networks, the $F_1$ scores are low, indicating that on real-world networks we loose essential information about the optimal percolation problem if the multiplex structure is not taken into account.

To provide a practical case study with an intuitive interpretation, we depict in Fig. 5 the solution of the optimal percolation problem on a multiplex network describing air transportation in the United States. SA identifies always 10 airports in the set of SNs of this network. There is a slight variability among different instances of the SA optimization, with a total of 14 distinct airports appearing in the structural set at least once over 100 SA instances. However, changes in the SN set from run to run mostly regard airports in the same geographical region. Overall, airports in the structural set are scattered homogeneously across the country, suggesting that the GMCC of the network mostly relies on hubs serving specific geographical regions, rather than global hubs in the entire transportation system. For instance, the probabilities that describe the membership of the airports to the set of SNs do not strictly follow the same order as that of the recorded flight traffics; nor merely the number of connections of the airports (not shown) is sufficient to determine the SNs.

## Discussion

In this paper, we studied the optimal percolation problem on multiplex networks. The problem regards the detection of the minimal set of nodes (i.e., the set of structural nodes, SNs) such that, if its members are removed from the network, the network is dismantled. The solution to the problem provides important information on the microscopic parts that should be maintained in a functional state to keep the overall system functioning, in a scenario of maximal stress. Our study focused mostly on the characterization of the SN sets of a given multiplex network in comparison with those found on the single-layer projections of the same multiplex, i.e., in a scenario where one "forgets" about the multiplex nature of the system. Our results demonstrate that, generally, multiplex networks have considerably smaller sets of SNs compared to the SN sets of their single-layer based network representations. The error committed when relying on single-layer representations of the multiplex does not regard only the size of the SN sets, but also the identity of the SNs. Both issues emerge in the analysis of synthetic network models, where edge overlap and/or interlayer degree–degree correlations seem to fully explain the amount of discrepancy between the SN set of a multiplex and the SN sets of its single-layer based representations. These issues are apparent also in many of the real-world multiplex networks we analyzed. Overall, we conclude that neglecting the multiplex structure of a network system subjected to maximal structural stress may result in significant inaccuracies about its robustness.

**Data availability**. Real multiplex networks analyzed in the paper have been constructed using data publicly available on the Web (see references in Table 1). The source code of the

implementation of the various algorithms used in the paper is available from the authors upon request.

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

## Acknowledgements

A.F. and F.R. acknowledge support from the US Army Research Office (W911NF-16-1-0104). F.R. acknowledges support from the National Science Foundation (Grant CMMI-1552487).

## Author contributions

All authors contributed to all aspects of this work.

## Additional information

**Competing interests:** The authors declare no competing financial interests.

