## [Peer Review File · Nature Communications]

Reviewers' comments:

Reviewer #1 (Remarks to the Author):

Summary:

The article focuses on understanding the problem of optimal percolation in multiplex networks. This is a topic that has not received a great deal of attention yet. The authors endeavour to establish the comparison between analysis of optimal percolation in the strict multi-layer context and simplified versions of the problem the only involve single layers (individual or composite). It is clear in this analysis that as the multi-layer feature of the problem is sharpened, the are significant changes to the situation, and those nodes that are identified solely on the basis of multi-layer networks are not easily identified as this is relaxed. The article addresses a timely and and relevant topic in terms of our growing understanding of multi-layer networks, and therefore I believe the work deserves attention.

General comments:

Being an extremal problem, it is not easy to expect an analytical framework to be available to solve the problem in a more formal way. Therefore, it is understandable that the authors approach the problem computationally.

My only concern with the current form of the manuscript is that the testing method to identify sensibility of node identification on the basis of the parameter α may not be enough to understand the problem. Looking at the practical example of American and Delta Airlines, one can clearly see that those structural nodes identified are in fact operational hubs of the airlines at regional levels. They play critical roles not just because of the volume of traffic, but also because small flights that are otherwise not served by larger planes are usually served from these locations. But this entangles once again node degrees and edge overlaps to some extent, something the authors acknowledge when they first present their method in the ms.

Recommendation:

My opinion is that the article tackles a good problem, and does a good job of trying to learn features of the problem, but I feel that it comes short in providing an explanation of the phenomena that can reach the mark of a qualitative explanation of the problem.

I would suggest that if the authors are able to expand on this point by: 1) an additional, or even a distinct, method of testing, and/or 2) a well constructed analysis of how the quantitative features can be qualitatively summarised, then I believe the article can be published in Nat. Comm.

General suggestion: Requires a good grammar check.

More specific comments:

1) The summary of results in abstract is rather vague "We find that the multilayer nature of these systems, and more precisely multiplex characteristics such as edge overlap and interlayer degree-degree correlation, profoundly changes the properties of the set of nodes identified as the solution of the optimal percolation problem."

2) Intro: the sentence "... forgetting about the effective co-existence of different types of interactions may lead to the emergence of rather different features..." cannot be correct. The fact that a researcher forgets about something cannot affect the situation under study; it affects the analysis of the situation.

3) page 2, column 1, paragraph 2, in the sentence "... of combining these numbers to assign a

single score to a node: ..." a first reference to a "score" for nodes is mentioned but no prior hint or context of this has been given before. It seems to come from the relation between this article, and the one by Grassberger (Clusella et al. PRL 117, 208301 (2016)), but not clear.

4) page 2, column 1, paragraph 2, in the sentence " We find that the results of the various algorithms are not particularly sensible to this choice, provided that a simple post-processing technique is applied to the set of SNs found by a given method." I suspect the authors meant to use the word "sensitive" rather than "sensible", i.e., meaning that the results do not change very much if the way a node is scored is changed.

5) In the same sentence as in 4), mention of a "post-processing technique" is made but not really clarified, nor is any preparation given to the reader about it. The next sentence may contain a clue about it, indicating that the nodes which do not increase the size of the GMCC beyond $N^{1/2}$ are removed. I am trying to interpret what I'm reading, but if my interpretation is correct, this should at least be written differently so that it is clearer what post-processing means.

6) Unless this is a journal rule, I would suggest that authors add the symbolic label of the quantity being presented to the axes of the plots. Personally, I don't mind the explained axes (I think they are useful to the quick reader), but think it clarifies for the detailed reader. So for instance, Fig 2B could have the vertical axis as 'relative error (epsilon)' making the connection between symbols in the text and in plots transparent.

7) Page 3, column 1, paragraph 1, In the sentence saying " The relative error associated with the aggregated network is instead larger...", I would take away the word "instead". Misleading.

Reviewer #2 (Remarks to the Author):

This work is devoted to the timely problem of the specifics of optimal percolation for multiplex networks. The phase transition in multiplex networks, associated with the emergence of a giant mutually connected component is discontinuous, which differs principally from uniplex networks, and so the optimisation problem is even more difficult (while it is already an NP-complex for uniplexes) and novel. The authors performed a comprehensive analysis of a set of optimisation algorithm, including simple and more precise while slow and practical only for small networks. Valuably, the analysis includes not only sizes of minimal destructing sets of vertices but also the lists of individual vertices in these sets. The authors also analyse the influence of structural correlations in multiplex networks, including correlations between edges from different layers (i.e., edge overlaps).

This work is interesting for many researchers, since the topic is one of the issues of great interest just now.

The weak point of presentation is that it is practically impossible to grasp the essence of the results from the abstract. In my opinion, it is not enough to only write in the abstract that multiplex organisation "profoundly changes the properties of " the minimal destructive set compared with the single-layer projection. The same could be said about conclusions, particularly, the key, last sentence.

In summary, I recommend that this work be accepted but the abstract and conclusions have to contain more clear message.

Reviewer #3 (Remarks to the Author):

This is an interesting clearly written paper, in which the authors point out and characterize the discrepancies that arise in optimal percolation when we regard a multiplex as simplex (single-layer) network. The process of optimally dismantling a multiplex networks relies on a small set of nodes who, collectively, break global connectivity in a coordinated manner. The attempt estimate

the size and identify these sets of key nodes while ignoring the multilayer structure of the system results in a great deal of errors, even using the most advanced technics for network dismantling, as is emphasized by the authors and clearly demonstrated by Table 1 and Figure 5. The observations presented in this paper are relevant and pertinent to the ongoing scientific discussion about the role and importance of multilayered systems in network science.

In my opinion this is a quality paper of significant interest to the field, which is suitable for publication in its present form.

First Reviewer

Summary: The article focuses on understanding the problem of optimal percolation in multiplex networks. This is a topic that has not received a great deal of attention yet. The authors endeavour to establish the comparison between analysis of optimal percolation in the strict multi-layer context and simplified versions of the problem the only involve single layers (individual or composite). It is clear in this analysis that as the multi-layer feature of the problem is sharpened, there are significant changes to the situation, and those nodes that are identified solely on the basis of multi-layer networks are not easily identified as this is relaxed. The article addresses a timely and relevant topic in terms of our growing understanding of multi-layer networks, and therefore I believe the work deserves attention.

We thank the reviewer for the positive feedback. In the following, we provide a point-to-point reply to all the concerns.

General comments:

1. Being an extremal problem, it is not easy to expect an analytical framework to be available to solve the problem in a more formal way. Therefore, it is understandable that the authors approach the problem computationally.

My only concern with the current form of the manuscript is that the testing method to identify sensibility of node identification on the basis of the parameter α may not be enough to understand the problem. Looking at the practical example of American and Delta Airlines, one can clearly see that those structural nodes identified are in fact operational hubs of the airlines at regional levels. They play critical roles not just because of the volume of traffic, but also because small flights that are otherwise not served by larger planes are usually served from these locations. But this entangles once again node degrees and edge overlaps to some extent, something the authors acknowledge when they first present their method in the ms.

Recommendation:

My opinion is that the article tackles a good problem, and does a good job of trying to learn features of the problem, but I feel that it comes short in providing an explanation of the phenomena that can reach the mark of a qualitative explanation of the problem.

I would suggest that if the authors are able to expand on this point by: 1) an additional, or even a distinct, method of testing, and/or 2) a well constructed analysis of how the quantitative features can be qualitatively summarised, then I believe the article can be published in Nat. Comm.

We agree with the interpretation provided by the reviewer about the air transportation multiplex studied in our manuscript. Some of these comments were (and still are) used to qualitatively describe figure 5, as we wrote: “Overall, airports in the structural set are

scattered homogeneously across the country, suggesting that the GMCC of the network mostly relies on hubs serving specific geographical regions, rather than global hubs in the entire transportation system.“

Regarding the comment on the testing method, we can state that the multiplex model consisting of two identical layers where nodes are relabelled with a certain probability is one of the simplest way of tuning the topology of a system from an isolated network to a multiplex network. The model has been used in previous literature, see for example Bianconi and Dorogovtsev, arXiv:1411.4160 (2014); Radicchi, Nat. Phys. 11, 597 (2015). In our manuscript, the model is used to address the main scientific question of the paper that is “What are the consequences of neglecting the multiplex nature of a network under an optimal percolation process?” We believe that the analysis presented in the paper provides sufficient evidence of the fact that the solution to the problem of optimal percolation in multiplex networks cannot be satisfactory approximated by solutions of the same problem in single-layer reductions/projections of the multiplex. In this respect, we do not see the need for any additional testing procedure in the current paper.

We see, however, the necessity for additional testing methods if the goal is to address deeper aspects of the optimal percolation problem. Our current analysis for instance doesn’t quantify the impact of multiplex features on the solution of the optimal percolation problem. Several properties can in principle be tested, e.g., degree distribution, degree correlation, edge overlap, geometric correlations. The importance of each of these features can be systematically studied with ad-hoc models. We feel, however, that such an extensive analysis may be better suited for future studies, as it goes well beyond the scope of the current paper.

In summary, we believe that “1) an additional, or even a distinct, method of testing” and “2) a well constructed analysis of how the quantitative features can be qualitatively summarised” are not required in the current paper. The main purpose of the current paper is to extend the problem (and algorithms aimed at solving the problem) of optimal percolation from uniplex to multiplex networks, and we believe that the current manuscript does already a good job in this respect. We would prefer to keep the focus of the manuscript on these aspects, leaving more specific analyses of the problem to future studies. We hope that the referee will agree with our point of view after reading the current reply, and the comments made by the other two reviewers.

General suggestion: Requires a good grammar check.
--

We corrected many grammatical errors. Also, we modified some sentences. Major changes to the text are highlighted in bold face in the revised version of the manuscript.

More specific comments:

1) The summary of results in abstract is rather vague “We find that the multilayer nature of these systems, and more precisely multiplex characteristics such as edge overlap and interlayer degree-degree correlation, profoundly changes the properties of the set of nodes identified as the solution of the optimal percolation problem.”

We expanded and modified the abstract.

2) Intro: the sentence "... forgetting about the effective co-existence of different types of interactions may lead to the emergence of rather different features" cannot be correct. The fact that a researcher forgets about something cannot affect the situation under study; it affects the analysis of the situation.

We corrected the sentence.

3) page 2, column 1, paragraph 2, in the sentence " of combining these numbers to assign a single score to a node: " a first reference to a "score" for nodes is mentioned but no prior hint or context of this has been given before. It seems to come from the relation between this article, and the one by Grassberger (Clusella et al. PRL 117, 208301 (2016)), but not clear.

We expanded the sentence.

4) page 2, column 1, paragraph 2, in the sentence " We find that the results of the various algorithms are not particularly sensible to this choice, provided that a simple post-processing technique is applied to the set of SNs found by a given method." I suspect the authors meant to use the word "sensitive" rather than "sensible", i.e., meaning that the results do not change very much if the way a node is scored is changed.

Thanks. We changed "sensible" to "sensitive"

5) In the same sentence as in 4), mention of a "post-processing technique" is made but not really clarified, nor is any preparation given to the reader about it. The next sentence may contain a clue about it, indicating that the nodes which do not increase the size of the GMCC beyond $N^{1/2}$ are removed. I am trying to interpret what I'm reading, but if my interpretation is correct, this should at least be written differently so that it is clearer what post-processing means.

We modified the sentence and added the corresponding reference.

6) Unless this is a journal rule, I would suggest that authors add the symbolic label of the quantity being presented to the axes of the plots. Personally, I don't mind the explained axes (I think they are useful to the quick reader), but think it clarifies for the detailed reader. So for instance, Fig 2B could have the vertical axis as relative error (epsilon) making the connection between symbols in the text and in plots transparent.

We prefer to keep the figure format as it is. This is just a stylistic choice. Further, we believe that a detailed reader should not have issues in finding details in the captions of the various figures.

7) Page 3, column 1, paragraph 1, In the sentence saying “ The relative error associated with the aggregated network is instead larger”, I would take away the word “instead”. Misleading.

We modified the sentence.

Second Reviewer

This work is devoted to the timely problem of the specifics of optimal percolation for multiplex networks. The phase transition in multiplex networks, associated with the emergence of a giant mutually connected component is discontinuous, which differs principally from uniplex networks, and so the optimisation problem is even more difficult (while it is already an NP-complex for uniplexes) and novel. The authors performed a comprehensive analysis of a set of optimisation algorithms, including simple and more precise while slow and practical only for small networks. Valuably, the analysis includes not only sizes of minimal destructing sets of vertices but also the lists of individual vertices in these sets. The authors also analyse the influence of structural correlations in multiplex networks, including correlations between edges from different layers (i.e., edge overlaps). This work is interesting for many researchers, since the topic is one of the issues of great interest just now. The weak point of presentation is that it is practically impossible to grasp the essence of the results from the abstract. In my opinion, it is not enough to only write in the abstract that multiplex organisation "profoundly changes the properties of " the minimal destructive set compared with the single-layer projection. The same could be said about conclusions, particularly, the key, last sentence. In summary, I recommend that this work be accepted but the abstract and conclusions have to contain more clear message.

We thank the reviewer for the very positive report. We expanded the abstract, and slightly modified the conclusions of the manuscript.

Third Reviewer

This is an interesting clearly written paper, in which the authors point out and characterize the discrepancies that arise in optimal percolation when we regard a multiplex as simplex (single-layer) network. The process of optimally dismantling a multiplex networks relies on a small set of nodes who, collectively, break global connectivity in a coordinated manner. The attempt estimate the size and identify these sets of key nodes while ignoring the multilayer structure of the system results in a great deal of errors, even using the most advanced technics for network dismantling, as is emphasized by the authors and clearly demonstrated by Table 1 and Figure 5. The observations presented in this paper are relevant and pertinent to the ongoing scientific discussion about the role and importance of multilayered systems in network science.

In my opinion this is a quality paper of significant interest to the field, which is suitable for publication in its present form.

We thank the reviewer for the very positive report.

List of changes

1. We rephrased/expanded the abstract and conclusions.
2. We performed changes in other parts of the text. To ease the re-review process, we highlighted in bold face major changes.
3. We modified manuscript, figures, and supplementary information to comply with the standards of the journal.

REVIEWERS' COMMENTS:

Reviewer #1 (Remarks to the Author):

I am satisfied with the authors comments on my review.

Reviewer #3 (Remarks to the Author):

The paper is in conditions to be published in my opinion.